

# Effect of metabolic syndrome on testosterone levels in patients with metastatic prostate cancer: a real-world retrospective study

Tao Zhuo[1], Xiangyue Yao[2], Yujie Mei[3], Hudie Yang[4], Abudukeyoumu Maimaitiyiming[1], Xin Huang[1], Zhuang Lei[1], Yujie Wang[1], Ning Tao[3] and Hengqing An[1]

[1] The First Affiliated Hospital, Xinjiang Medical University, Urumqi, Xinjiang, China
[2] The Fifth Affiliated Hospital, Xinjiang Medical University, Urumqi, Xinjiang, China
[3] College of Public Health, Xinjiang Medical University, Urumqi, Xinjiang, China
[4] The Second Affiliated Hospital, Xinjiang Medical University, Urumqi, Xinjiang, China

Corresponding authors
Ning Tao, 38518412@qq.com
Hengqing An, 9269735@qq.com

## ABSTRACT

**Background.** Metabolic syndrome (MetS) has been shown to have a negative impact on prostate cancer (PCa). However, there is limited research on the effects of MetS on testosterone levels in metastatic prostate cancer (mPCa).

**Objective.** This study aims to investigate the influence of MetS, its individual components, and composite metabolic score on the prognosis of mPCa patients, as well as the impact on testosterone levels. Additionally, it seeks to identify MetS-related risk factors that could impact the time of decline in testosterone levels among mPCa patients.

**Methods.** A total of 212 patients with mPCa were included in the study. The study included 94 patients in the Non-MetS group and 118 patients in the combined MetS group. To analyze the relationship between MetS and testosterone levels in patients with mPCa. Additionally, the study aimed to identify independent risk factors that affect the time for testosterone levels decline through multifactor logistic regression analysis. Survival curves were plotted by the Kaplan-Meier method.

**Results.** Compared to the Non-MetS group, the combined MetS group had a higher proportion of patients with high tumor burden, T stage ≥ 4, and Gleason score ≥ 8 points ($P < 0.05$). Patients in the combined MetS group also had higher lowest testosterone values and it took longer for their testosterone to reach the lowest level ($P < 0.05$). The median progression-free survival (PFS) time for patients in the Non-MetS group was 21 months, while for those in the combined MetS group it was 18 months ($P = 0.001$). Additionally, the median overall survival (OS) time for the Non-MetS group was 62 months, whereas for the combined MetS group it was 38 months ($P < 0.001$). The median PFS for patients with a composite metabolic score of 0–2 points was 21 months, 3 points was 18 months, and 4–5 points was 15 months ($P = 0.002$). The median OS was 62 months, 42 months, and 29 months respectively ($P < 0.001$). MetS was found to be an independent risk factor for testosterone levels falling to the lowest value for more than 6 months. The risk of testosterone levels falling to the lowest value for more than 6 months in patients with MetS was 2.157 times higher than that of patients with Non-MetS group ($P = 0.031$). Patients with hyperglycemia had a significantly higher lowest values of testosterone ($P = 0.015$). Additionally, patients with

![PeerJ]

a BMI $\geq 25$ kg/m$^2$ exhibited lower initial testosterone levels ($P = 0.007$). Furthermore, patients with TG $\geq 1.7$ mmol/L experienced a longer time for testosterone levels to drop to the nadir ($P = 0.023$). The lowest value of testosterone in the group with a composite metabolic score of 3 or 4–5 was higher than that in the 0–2 group, and the time required for testosterone levels to decrease to the lowest value was also longer ($P < 0.05$).

**Conclusion**. When monitoring testosterone levels in mPCa patients, it is important to consider the impact of MetS and its components, and make timely adjustments to individualized treatment strategies.

## INTRODUCTION

According to global cancer data, prostate cancer (PCa) is the fourth most common type of cancer, accounting for 7.3% of new cases, and it is the second most common solid tumor in men worldwide (*Sung et al., 2021*). The incidence of PCa varies across different regions, with higher rates observed in Europe and the United States compared to Asia. In recent years, there has been an increasing incidence of PCa in China (*Gandaglia et al., 2021*; *He et al., 2022*). Metastatic prostate cancer (mPCa) is a critical stage that significantly impacts patient prognosis. Studies have shown that the 5-year survival rate for mPCa is only 31% (*He et al., 2022*). Additionally, the incidence of middle- and late-stage PCa in China is higher than that in Europe and the United States (*Miller et al., 2022*).

Age, family history, and genetic susceptibility are established risk factors for PCa. Additionally, several factors including tumor stage, tumor burden, and Gleason score have been found to be strongly associated with the prognosis of PCa patients. Metabolic syndrome (MetS) is a pathological condition characterized by abdominal obesity, insulin resistance, hypertension, and hyperlipidemia, as defined by the World Health Organization. The diagnostic criteria for MetS may vary slightly among different countries or medical organizations, but the fundamental features remain the same. In 2004, the association between MetS and the risk of developing PCa was first observed (*Laukkanen et al., 2004*). Since then, several studies have confirmed that MetS has a negative impact on both the occurrence and prognosis of PCa (*Hernández-Pérez et al., 2022*; *Gacci et al., 2017*). *Porretti et al. (2018)* conducted a study and suggested that this adverse effect may be caused by MetS influencing the epigenetic regulation of PCa. Androgen deprivation therapy (ADT) has been a significant treatment approach for mPCa since 1941. Its goal is to regulate testosterone levels in PCa patients, keeping them below castration levels. In the follow-up of PCa patients, monitoring testosterone levels is just as crucial as monitoring prostate-specific antigen (PSA) levels (*Schulman et al., 2010*). While some studies have explored the association between MetS and testosterone, there is a lack of relevant reports in the mPCa population. This study aims to investigate the influence of MetS, its individual components, and composite metabolic score on the prognosis of mPCa patients, as well as

the impact on testosterone levels before and after treatment. Additionally, the study aimed to identify potential risk factors that could influence the time of testosterone levels decline in mPCa patients.

## DATA AND METHODS

### Subjects and methods of study

The clinical data of 212 patients diagnosed with mPCa were retrospectively analyzed at the Urology Center of the First Affiliated Hospital of Xinjiang Medical University. The analysis included 94 cases in the Non-MetS group and 118 cases in the combined MetS group. The study examined the effects of MetS, each component of MetS, and the composite metabolic score on prognosis, initial testosterone levels, lowest testosterone values, and the time it took for testosterone levels to decline to the lowest value in patients with mPCa. Additionally, multi-factor logistic regression analysis was conducted to identify independent risk factors that affect the time it takes for testosterone levels to decline. Survival curves were plotted by the Kaplan–Meier method.

### Follow-up and observation indicators

Clinical data collected for this study encompassed various factors including age (years), ethnicity, smoking history, drinking history, hypertension history, hyperglycemia history, T stage, nerve invasion, visceral metastasis, Gleason score (the highest in the pathology report), initial PSA levels (ng/mL), initial testosterone levels (nmol/L), tumor burden, blood pressure (mmHg), body mass index (BMI, kg/m$^2$), fasting blood glucose (FBG, mmol/L), triglyceride (TG, mmol/L), high density lipoprotein (HDL, mmol/L), lowest value of testosterone within 1 year of treatment (nmol/L), and time of testosterone levels decline to the lowest value (months). The time to progress to castration resistant prostate cancer (CRPC) is defined as the duration from the start of endocrine therapy to the diagnosis of CRPC. Overall survival (OS) is measured from the diagnosis of mPCa and the initiation of regular endocrine therapy until death from any cause or the length of the last follow-up. Before receiving endocrine therapy, all mPCa patients underwent MetS assessment. All patients diagnosed with MetS undergo regular treatment and rehabilitation exercises overseen by specialized healthcare professionals. All patients with mPCa in this research were prescribed comparable endocrine therapy plans, predominantly consisting of goserelin in conjunction with abiraterone.

### Related definitions

(1) Initial PSA levels and initial testosterone levels: These are the serum total PSA levels and total testosterone levels measured when the patient was diagnosed but had not received any treatment yet.

(2) Testosterone lowest value: Lowest value of testosterone within 1 year of treatment.

(3) Time to drop to the lowest value of testosterone: This indicates the time interval from the start of treatment to the point at which the lowest value of testosterone was recorded during the follow-up.

(4) The definition of MetS is based on relevant international guidelines and China's Type 2 Diabetes Prevention and Treatment Guidelines (2020 Edition) (*Saklayen, 2018*).

The diagnosis of MetS requires the presence of at least the following three criteria: BMI ≥ 25 kg/m$^2$; hyperglycemia: FBG ≥ 6.1 mmol/L or individuals with blood sugar ≥ 7.8 mmol/L 2 h after glucose loading and/or those who have already been diagnosed with diabetes and are receiving treatment; hypertension: individuals with blood pressure ≥ 130/85 mmHg (1 mmHg = 0.133 kPa) and/or those who have already been diagnosed with hypertension and are receiving treatment; fasting TG ≥ 1.70 mmol/L; fasting HDL <l.04 mmol/L. Each of these criteria is assigned a value of 1 point and individuals are categorized into three groups based on the composite metabolic score (0–2 points, 3 points, and 4–5 points). MetS is diagnosed when the composite metabolic score is ≥3 points. (5) Dyslipidemia is defined as having Fasting TG ≥ 1.70 mmol/L and/or fasting HDL <1.04 mmol/L.

## Inclusion and exclusion criteria

Inclusion criteria: (1) Patients who have been clearly diagnosed with mPCa based on pathological and related auxiliary examinations. (2) Patients who have not received any endocrine treatment in the past, as this may affect the observation indicators. (3) Patients who are diagnosed with mPCa in our hospital and have started receiving regular endocrine treatment. (4) Patients with complete clinical data.

Exclusion criteria: (1) Patients who had been previously diagnosed with PCa and received treatment; (2) Patients who received other treatments before progressing to CRPC, such as palliative surgery, radiotherapy, molecular targeted therapy, or docetaxel chemotherapy; (3) Patients who had other malignant tumors at the time of their initial diagnosis; (4) Patients who had other serious diseases; (5) Patients who were lost to follow-up or had missing data during the follow-up process; (6) Patients who have used 5α reductase inhibitors, testosterone replacement drugs, and other external factors that may affect testosterone; (7) those who have undergone surgical castration.

A total of 212 mPCa patients were included in this study based on strict inclusion and exclusion criteria. The study screening flowchart is depicted in Fig. 1.

## Ethical review

This study is retrospective, and the subjects studied were informed or supplemented with the purpose of the study, the content of the study, and the potential patient benefits and risks of the study, and the patients themselves or their families were informed by telephone or signed the informed consent in writing, and the study complied with the principles of the Declaration of Helsinki and the principles of the Medical Ethics Committee of the First Affiliated Hospital of Xinjiang Medical University (No.20220308-166).

## Statistical methods

Data analysis was conducted using SPSS 25.0 software. Measurement information was described using $\bar{x} \pm s$ or M (P25, P75). Two-group comparisons were performed using t-tests or non-parametric tests. Multiple comparisons were conducted using one-way ANOVA. Count information was described using rates, and inter-group comparisons were performed using $\chi^2$ tests. Hierarchical information was analyzed using rank-sum tests. Variables with statistically significant differences were included in a multi-factor logistic regression model to identify independent risk factors affecting the time of testosterone

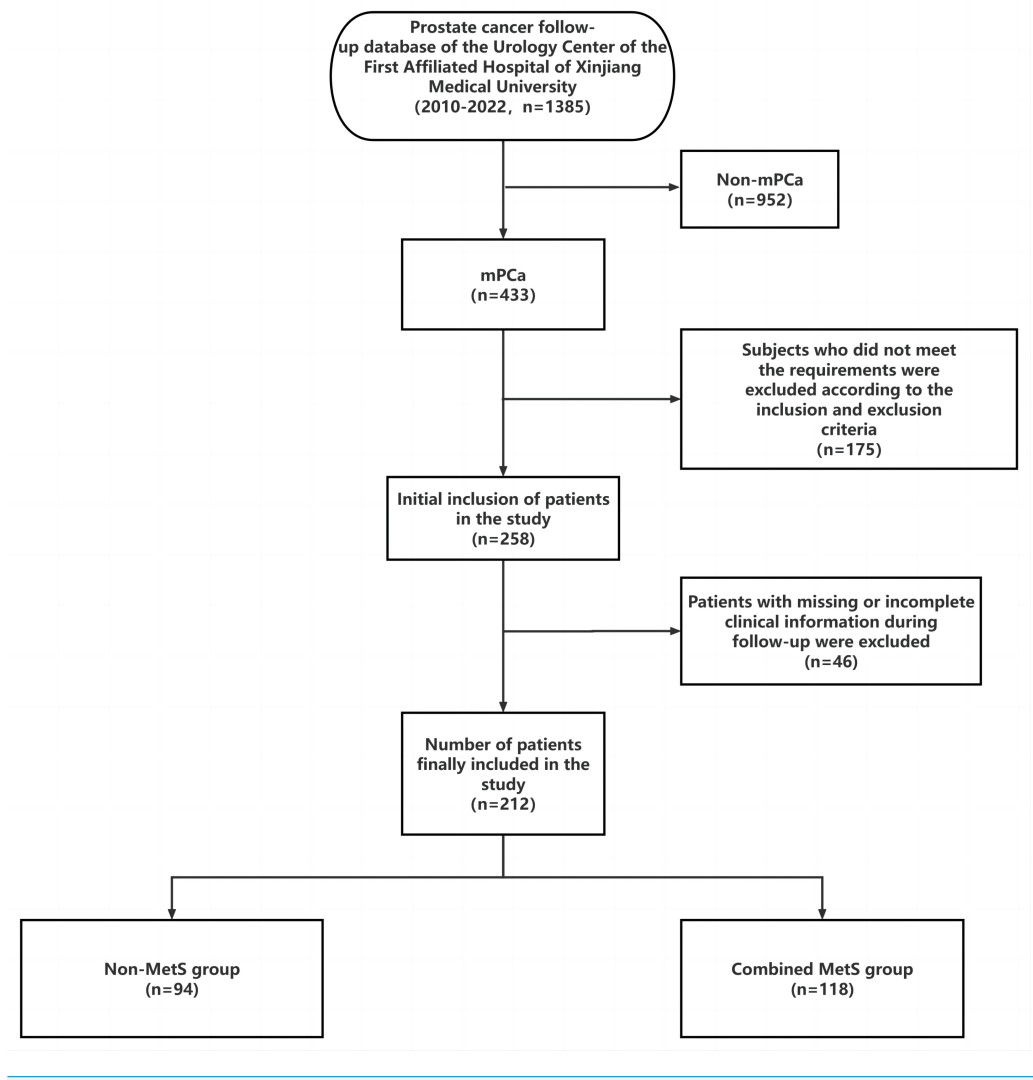

**Figure 1   Research flow chart.**

levels decline. Survival curves were plotted by the Kaplan–Meier method. GraphPad Prism 9.4.1 software was used for visualization. The test level was set at $\alpha = 0.05$.

## RESULTS

### Comparison of clinical data between patients in the Non-MetS and combined MetS groups

There was no statistically significant difference between the two groups in terms of age, ethnicity, smoking, drinking, nerve invasion, visceral metastasis, initial PSA levels, and initial testosterone levels ($P > 0.05$). However, when comparing the two groups, there were statistically significant difference in tumor burden, T-stage, Gleason score, lowest value of testosterone, and time of testosterone levels decline to the lowest value ($P < 0.05$). The combined MetS group had a higher proportion of patients with high tumor burden,

**Table 1  Comparison of clinical data between patients in the Non-MetS and combined MetS groups.**

| Groups | Non-MetS group (94 cases) n (%)/M (P25, P75) | Combined MetS group (118 cases) n (%)/M (P25, P75) | t/z/χ² | P |
|---|---|---|---|---|
| Age (years) | 70.78 ± 5.809 | 72.28 ± 7.637 | −1.627 | 0.105 |
| Ethnicity | | | 0.218 | 0.640 |
| Han ethnic | 54 (57.4) | 64 (54.2) | | |
| Others | 40 (42.6) | 54 (45.8) | | |
| Smoking | | | 0.026 | 0.872 |
| No | 56 (59.6) | 69 (58.5) | | |
| Yes | 38 (40.4) | 49 (41.5) | | |
| Drinking | | | 0.055 | 0.814 |
| No | 62 (66.0) | 76 (64.4) | | |
| Yes | 32 (34.0) | 42 (35.6) | | |
| Nerve invasion | | | 0.001 | 0.973 |
| No | 52 (55.3) | 65 (55.1) | | |
| Yes | 42 (44.7) | 53 (44.9) | | |
| Visceral metastasis | | | 0.219 | 0.640 |
| No | 75 (79.8) | 91 (77.1) | | |
| Yes | 19 (20.2) | 27 (22.9) | | |
| Tumor burden | | | 7.385 | 0.007 |
| Low | 45 (47.9) | 35 (29.7) | | |
| High | 49 (52.1) | 83 (70.3) | | |
| T stage | | | 7.719 | 0.005 |
| <4 | 48 (51.1) | 38 (32.2) | | |
| ≥4 | 46 (48.9) | 80 (67.8) | | |
| Gleason score (score) | | | 8.255 | 0.004 |
| <8 | 20 (21.3) | 9 (7.6) | | |
| ≥8 | 74 (78.7) | 109 (92.4) | | |
| Initial PSA levels (ng/mL) | | | 0.142 | 0.706 |
| <100 | 28 (29.8) | 38 (32.2) | | |
| ≥100 | 66 (70.2) | 80 (67.8) | | |
| Initial testosterone levels (nmol/L) | 13.90 ± 5.448 | 12.61 ± 5.078 | 1.786 | 0.076 |
| Lowest value of testosterone (nmol/L) | 0.08 (0.03, 0.18) | 0.25 (0.07, 0.61) | −4.287 | <0.001 |
| Time of testosterone decline to the lowest value (months) | 5.00 (3.00, 6.00) | 6.00 (4.75, 9.00) | −4.089 | <0.001 |

T-stage ≥ 4, and Gleason score ≥ 8. Additionally, patients in the mPCa combined MetS group had higher lowest value of testosterone and longer time of testosterone levels decline to the lowest value compared to the Non-MetS group. Please refer to Table 1 for more details.
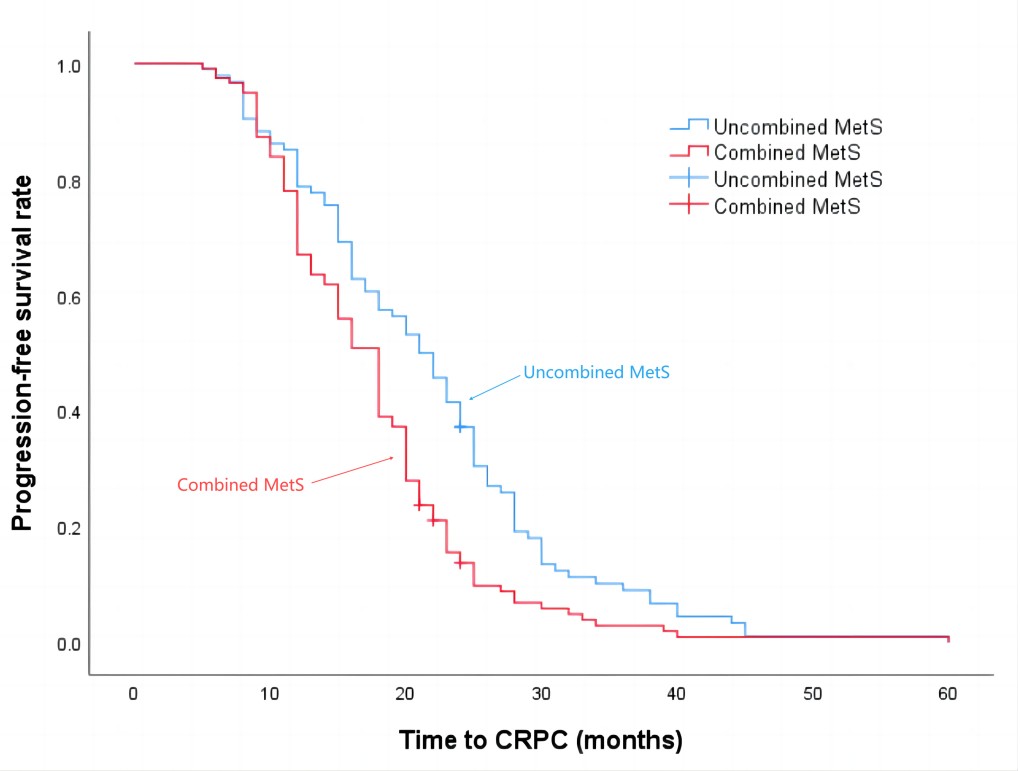

**Figure 2** Time to progression to CRPC in patients in the Non-MetS and combined MetS groups.

## Progression-free survival and overall survival are shorter in patients with MetS

The median progression-free survival time for patients in the Non-MetS group was 21 months, while for those in the combined MetS group it was 18 months ($P = 0.001$), the details are shown in Fig. 2. Additionally, the median survival time for the Non-MetS group was 62 months, whereas for the combined MetS group it was 38 months ($P < 0.001$), the details are shown in Fig. 3.

## A higher composite metabolic score is associated with shorter progression-free survival time and overall survival time in patients

The median progression-free survival time for patients with a composite metabolic score of 0–2 points was 21 months, 3 points was 18 months, and 4–5 points was 15 months ($P = 0.002$), the details are shown in Fig. 4. The median overall survival time was 62 months, 42 months, and 29 months respectively ($P < 0.001$), the details are shown in Fig. 5.

## Effect of MetS components on testosterone in patients with mPCa

Compared to patients without hyperglycemia, patients with hyperglycemia exhibited higher lowest value of testosterone ($P = 0.015$). Similarly, patients with a BMI $\geq 25$ kg/m$^2$ had lower initial testosterone levels compared to patients with a BMI $< 25$ kg/m$^2$ ($P = 0.007$).

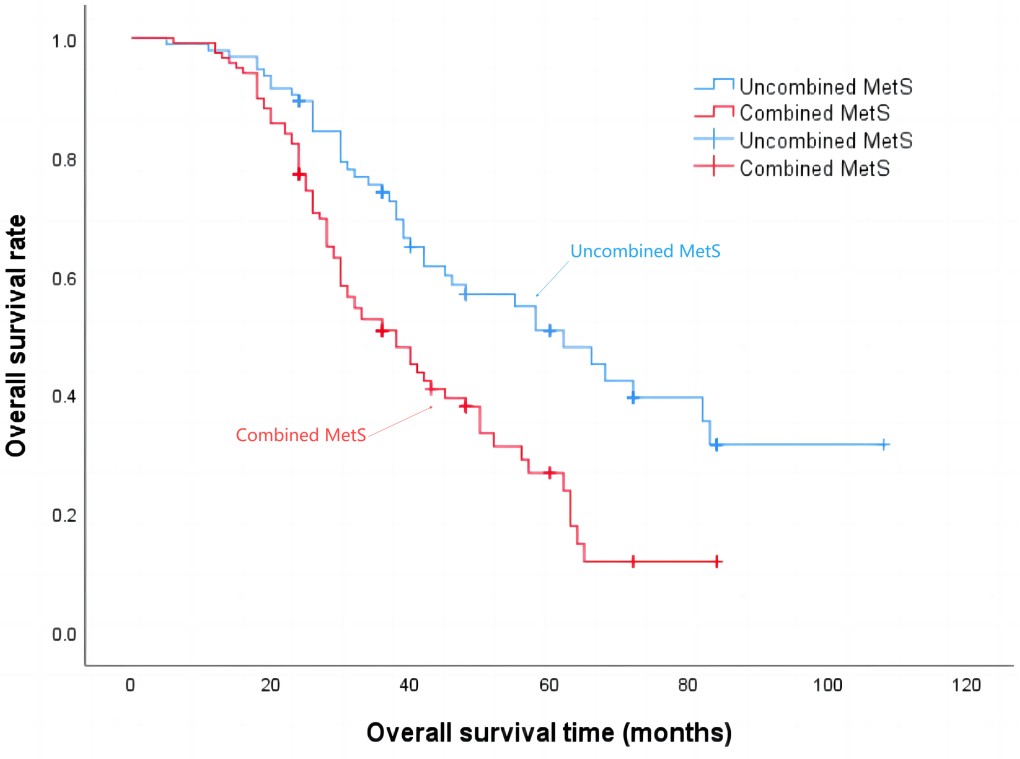

**Figure 3** **Overall survival time of patients in the Non-MetS and combined MetS groups.**

Patients with TG ≥ 1.7 mmol/L took longer time for testosterone levels to reach the lowest value compared to patients with TG <1.7 mmol/L ($P = 0.023$). No significant statistical difference was observed between the remaining groups ($P > 0.05$). Please refer to Tables 2A–2C for further details.

## MetS is an independent risk factor that influences the timing of testosterone decline

Compared to the group where the time for testosterone levels to reach the lowest value is ≤6 months, the group where the time for testosterone levels to reach the lowest value is >6 months has a higher proportion of patients with Gleason score ≥8 points, TG ≥1.7 mmol/L, and combined MetS ($P < 0.05$). Additionally, the patients in the group where the time for testosterone levels to reach the lowest value is >6 months also had higher FBG, and this difference was found to be statistically significant ($P < 0.05$). The statistically significant results were included in the multifactorial logistic regression analysis. It was determined that mPCa combined with MetS was an independent risk factor for the time it took for testosterone levels to decline to nadir for more than 6 months. The patients in the mPCa combined with MetS group had a 2.157 times higher risk of testosterone levels decline to nadir for more than 6 months compared to the patients in the Non-MetS group ($P = 0.031$). Please refer to Tables 3–4 for further details.

Zhuo et al. (2024), *PeerJ*, DOI 10.7717/peerj.17823

**Table 2    Effect of MetS components on testosterone in patients with mPCa.**

**2A**

| Groups | No hypertension (92 cases) | Hypertension (120 cases) | t/z | P | No hyperglycemia (122 cases) | Hyperglycemia (90 cases) | t/z | P |
|---|---|---|---|---|---|---|---|---|
| Initial testosterone levels (nmol/L) | 13.26 ± 5.016 | 13.12 ± 5.480 | 0.194 | 0.846 | 13.47 ± 5.514 | 12.79 ± 4.928 | 0.936 | 0.350 |
| Lowest value of testosterone (nmol/L) | 0.10 (0.05, 0.41) | 0.18 (0.06, 0.55) | −1.464 | 0.143 | 0.10 (0.05, 0.37) | 0.20 (0.06, 0.65) | −2.427 | 0.015 |
| Time of testosterone decline to the lowest value (months) | 5.50 (4.00, 7.00) | 5.00 (4.00, 8.00) | −0.001 | 0.999 | 5.00 (4.00, 7.00) | 6.00 (4.00, 8.00) | −1.825 | 0.068 |

**2B**

| Groups | TG < 1.7 mmol/L (133 cases) | TG ≥ 1.7 mmol/L (79 cases) | t/z | P | HDL < 1.04 mmol/L (125 cases) | HDL ≥ 1.04 mmol/L (87 cases) | t/z | P |
|---|---|---|---|---|---|---|---|---|
| Initial testosterone levels (nmol/L) | 13.29 ± 5.205 | 13.00 ± 5.412 | 0.387 | 0.699 | 12.99 ± 5.516 | 13.45 ± 4.920 | −0.627 | 0.531 |
| Lowest value of testosterone (nmol/L) | 0.12 (0.05, 0.53) | 0.13 (0.06, 0.53) | −0.793 | 0.428 | 0.15 (0.06, 0.56) | 0.11 (0.04, 0.45) | −1.704 | 0.088 |
| Time of testosterone decline to the lowest value (months) | 5.00 (4.00, 7.00) | 6.00 (5.00, 8.00) | −2.281 | 0.023 | 5.00 (4.00, 8.00) | 6.00 (4.00, 7.00) | −0.193 | 0.847 |

**2C**

| Groups | BMI < 25 kg/m² (112 cases) | BMI ≥ 25 kg/m² (100 cases) | t/z | P | No dyslipidemia (58 cases) | Dyslipidemia (154 cases) | t/z | P |
|---|---|---|---|---|---|---|---|---|
| Initial testosterone levels (nmol/L) | 14.10 ± 5.270 | 12.15 ± 5.105 | 2.740 | 0.007 | 13.25 ± 5.155 | 13.15 ± 5.332 | 0.126 | 0.900 |
| Lowest value of testosterone (nmol/L) | 0.11 (0.04, 0.49) | 0.14 (0.06, 0.55) | −1.496 | 0.135 | 0.11 (0.04, 0.29) | 0.14 (0.06, 0.55) | −1.707 | 0.088 |
| Time of testosterone decline to the lowest value (months) | 5.00 (4.00, 7.00) | 6.00 (4.00, 8.00) | −1.311 | 0.190 | 6.00 (4.00, 7.00) | 5.00 (4.00, 8.00) | −0.540 | 0.589 |

**Notes.**
Describe with $\bar{x} \pm s$ or M (P25, P75).

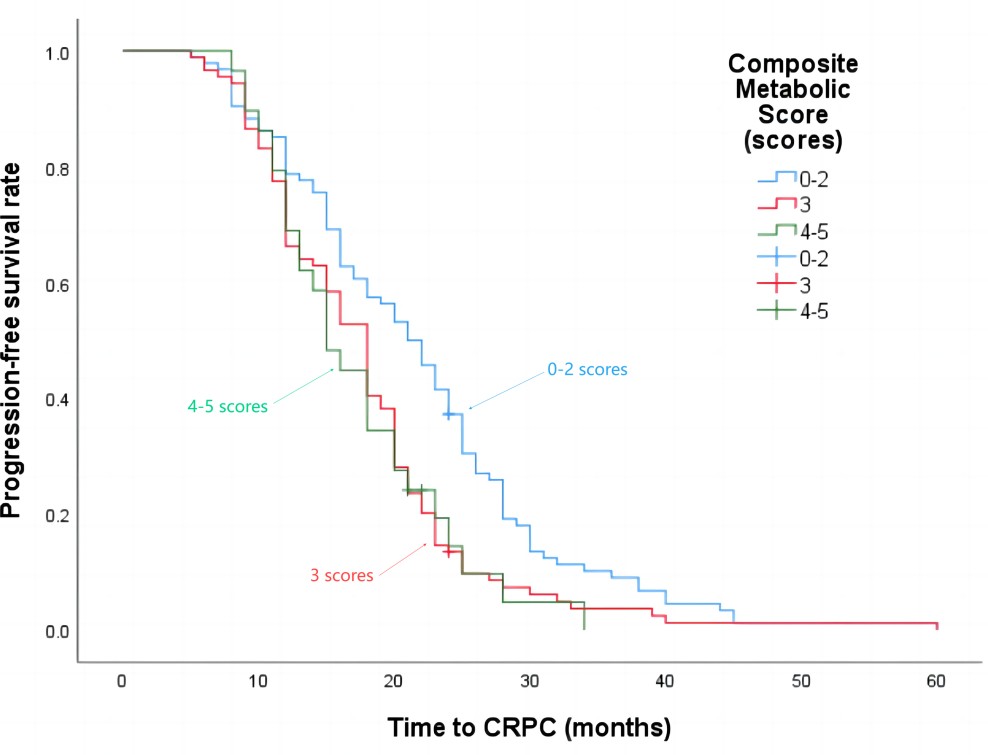

**Figure 4** Progression-free survival time in patients with different composite metabolic scores.

**Table 3  Single factor analysis affecting the time of testosterone decline to the lowest value.**

| Groups | Time to testosterone levels decline to the lowest value ≤ 6 months group (138 cases) n (%)/M (P25, P75) | Time to testosterone levels decline to the lowest value > 6 months group (74 cases) n (%)/M (P25, P75) | $z/\chi^2$ | P |
|---|---|---|---|---|
| Gleason score (score) | | | 4.614 | 0.032 |
| <8 | 24 (17.4) | 5 (6.8) | | |
| ≥8 | 114 (82.6) | 69 (93.2) | | |
| FBG mmol/L | 5.40 (4.93, 6.64) | 6.09 (5.07, 7.23) | −1.995 | 0.046 |
| TG mmol/L | | | 4.895 | 0.027 |
| <1.7 | 94 (68.1) | 39 (52.7) | | |
| ≥1.7 | 44 (31.9) | 35 (47.3) | | |
| MetS | | | 11.735 | 0.001 |
| No | 73 (52.9) | 21 (28.4) | | |
| Yes | 65 (47.1) | 53 (71.6) | | |

**Notes.**
Only statistically significant factors are listed.

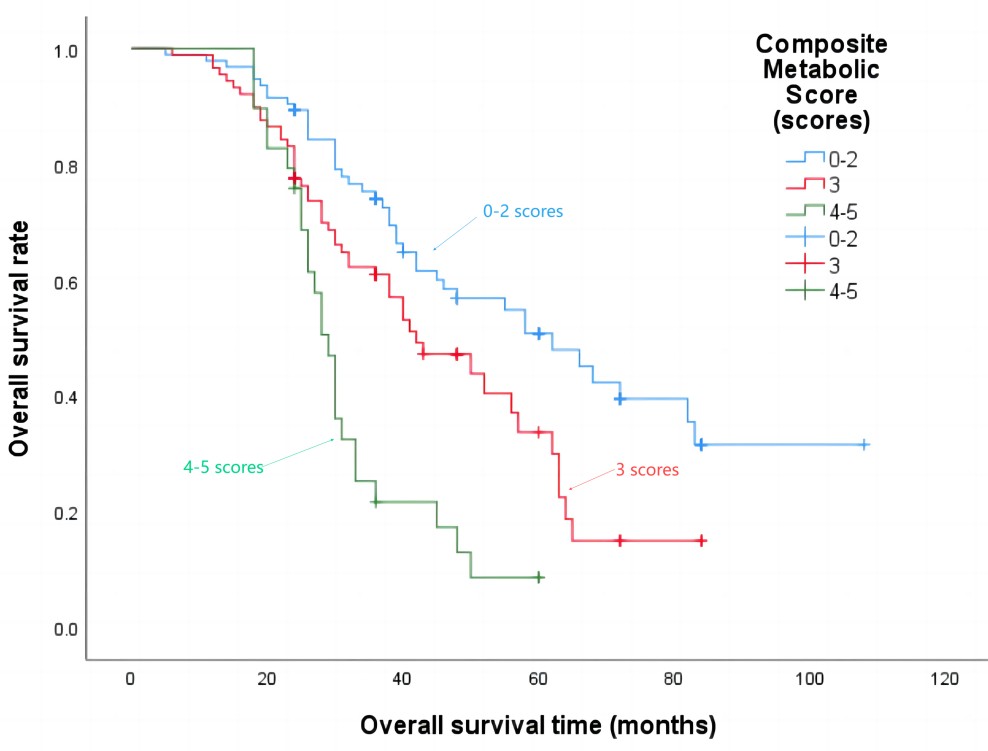

**Figure 5** Overall survival time in patients with different composite metabolic scores.

**Table 4  Multifactorial logistic regression analysis affecting the time of testosterone decline to the lowest value.**

| Variables | Regression coefficient | Standard error | *Wald $\chi^2$* | Exp (B) | 95% CI | *P* |
|---|---|---|---|---|---|---|
| Gleason score ≥ 8 score | 0.852 | 0.531 | 2.576 | 2.343 | 0.828–6.628 | 0.108 |
| FBG mmol/L | 0.051 | 0.080 | 0.407 | 1.052 | 0.900–1.231 | 0.523 |
| TG ≥ 1.7mmol/L | 0.326 | 0.333 | 0.957 | 1.385 | 0.721–2.660 | 0.328 |
| MetS-Yes | 0.769 | 0.355 | 4.677 | 2.157 | 1.075–4.330 | 0.031 |

## Effect of different composite metabolic scores on testosterone in patients with mPCa

There was a statistical difference observed among the different compound metabolic score groups (0–2, 3, and 4–5) when comparing lowest value of testosterone ($P = 0.006$) and time of testosterone levels decline to the lowest value ($P < 0.001$). However, there was no significant difference in the comparison of initial testosterone levels ($P = 0.149$). Subgroups with composite metabolic scores of 3 or 4–5 had higher lowest value of testosterone and longer time of testosterone levels decline to the lowest value compared to subgroups 0–2. However, there were no significant differences in lowest value of testosterone and time of

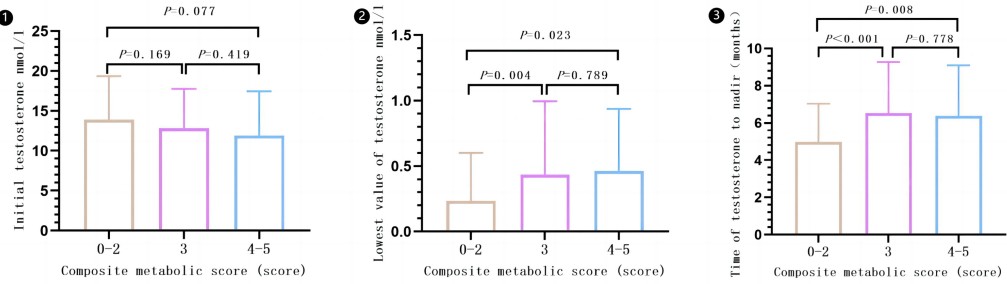

**Figure 6** **Effect of different composite metabolic scores on testosterone in patients with mPCa.**

testosterone levels decline to the lowest value between subgroups 3 and 4–5. Please refer to
Fig. 6 for more details (Results are expressed in $\overline{x} \pm s$).

## DISCUSSION

With the significant increase in human life expectancy, the prevalence of MetS has also
risen, affecting approximately 25% of adults worldwide (*Lotti et al., 2021*). MetS and its
related factors, including obesity, insulin resistance, impaired glucose tolerance, elevated
triglycerides and cholesterol, and reduced high-density lipoprotein, have been found to be
associated with reduced testosterone levels (*Berg & Miner, 2020*; *Dimopoulou et al., 2018*).
Testosterone is a crucial steroid hormone in the human body, playing a significant role in
male growth and development, as well as enhancing sexual desire, strength, reproductive
function, immune function, and fighting osteoporosis (*Kutlikova et al., 2020*). It also
regulates social stress (*Kutlikova et al., 2020*). Numerous studies have demonstrated
that individuals with low serum testosterone levels are at a higher risk of developing
PCa (*Mearini et al., 2013*). Additionally, PCa patients with low initial serum testosterone
levels tend to have higher levels of PSA, advanced tumor stage, and higher Gleason
score (*Bart et al., 2022*; *Gan et al., 2022*). Adverse prognostic factors such as extraprostatic
invasion, positive surgical margins, and postoperative pathological upgrading are also
associated with low initial serum testosterone levels (*Bart et al., 2022*; *Gan et al., 2022*). The
interaction between MetS and testosterone levels may be one of the potential mechanisms
underlying the pathogenesis and poor prognosis of PCa. However, there is a scarcity of
clinical studies investigating the effect of MetS on testosterone levels in patients with PCa.
This study conducted a preliminary exploration of the impact of MetS on testosterone
levels in patients with mPCa. Additionally, it analyzed the effect of MetS on the time of
testosterone levels decline in mPCa patients.

MetS is a crucial factor that impacts the health of middle-aged and elderly individuals. Its
connection with cancer has also been extensively studied. Existing literature has highlighted
the link between MetS and various types of cancer, including endometrial cancer, colorectal
cancer, gastric cancer, liver cancer, bladder cancer, and prostate cancer (*Belladelli, Montorsi
& Martini, 2022*). A study conducted on the Mexican population revealed that MetS is
linked to a higher occurrence of PCa (*Hernández-Pérez et al., 2022*). Risk factors such

as dyslipidemia, hypertension, and obesity contribute to an increased incidence of PCa (*Hernández-Pérez et al., 2022*). Additionally, obesity is found to be independently associated with a Gleason score of ≥8 (*Hernández-Pérez et al., 2022*). *Lebdai et al. (2018)* also discovered that MetS is an independent risk factor for positive margins in the radical resection of PCa (OR = 1.5, $P = 0.035$). MetS is also linked to the International Society of Urological Pathology (ISUP) grading system for radical resection specimens, with a close association to pathological grade ≥ 4 (OR = 2.0, $P = 0.044$) (*Lebdai et al., 2018*). Additionally, low levels of HDL are correlated with locally advanced PCa (*Lebdai et al., 2018*). A meta-analysis revealed a significant association between MetS and the incidence of PCa (OR = 1.17, $P = 0.04$), as well as high-grade PCa (OR = 1.89, $P < 0.0001$) (*Gacci et al., 2017*). Furthermore, MetS is significantly linked to a Gleason score of ≥8, seminal vesicle invasion, extraprostatic invasion, positive surgical margins, and biochemical recurrence (*Gacci et al., 2017*). Our findings indicate that there is a higher occurrence of high tumor burden, T stage ≥4, and Gleason score ≥8 in the combined MetS group, which aligns with previous research. These results suggest that mPCa combined with MetS exhibits a more aggressive nature. However, a recent study conducted on African-Caribbean men did not find a significant association between MetS and adverse pathological features or risk of biochemical recurrence (*Lefebvre et al., 2022*). It is important to note that the relationship between MetS and PCa may be influenced by various factors and different populations. To obtain a more objective and accurate correlation between the two, it is essential to conduct comprehensive research on a global scale, involving multiple regions and populations. This study also observed that patients with MetS experienced shorter progression-free survival and overall survival times. Consistent results were reported in the study conducted by *Polesel et al. (2016)*. Previous studies have indicated a negative correlation between MetS and levels of testosterone and sex hormone binding globulin (*Lotti et al., 2021*; *Cunningham, 2015*). In this study, there was no significant difference in the initial testosterone levels between the Non-MetS group and the combined MetS group, which could be attributed to the limited sample size. However, numerous studies have consistently demonstrated a strong association between low initial serum testosterone levels and the occurrence, progression, and poor prognosis of PCa (*Rodriguez, Pastuszak & Khera, 2018*). Furthermore, it has been observed that PCa patients with lower testosterone concentrations at castration level tend to have longer survival and progression time to castration-resistant prostate cancer (*Shiota et al., 2016*). Additionally, a rapid decline in testosterone to the lowest concentration during endocrine therapy has also been found to be beneficial for the prognosis (*Shiota et al., 2016*). In our study, we observed that mPCa patients with MetS had higher lowest testosterone values after treatment and a longer time to decline to the lowest testosterone value compared to the Non-MetS group. Additionally, we found a correlation between Gleason score ≥ 8 points, TG ≥ 1.7 mmol/L, fasting blood glucose, MetS, and the time it takes for testosterone levels to drop to the lowest value for more than 6 months. Importantly, MetS is considered an independent risk factor for testosterone levels declining to a nadir for more than 6 months. MetS may impact the aggressive characteristics and prognosis of mPCa patients by influencing testosterone levels prior to and after treatment, as well as the timing of testosterone levels decline. However, the specific mechanism of interaction between MetS

and testosterone levels in mPCa patients remains unclear. Further investigation into the correlation between these two factors and exploration of the underlying mechanism could contribute to personalized treatment strategies and improved prognosis for mPCa patients.

In addition to examining the overall impact of MetS on testosterone levels in patients with mPCa, we also investigated the effects of the individual components of MetS. Obesity and diabetes mellitus (DM) are prevalent in the population. Numerous studies have demonstrated a strong association between obesity, DM, and decreased levels of serum testosterone. Our study discovered that patients with a BMI ≥25 kg/m$^2$ exhibited lower initial testosterone levels, which aligns with previous research findings. Testosterone has been found to have a positive impact on body composition. In men, it has been observed that testosterone levels decrease as weight gain occurs. This correlation is influenced by various intricate interactions involving obesity, insulin resistance, and the hypothalamic-pituitary-testicular axis (*Umapathysivam, Grossmann & Wittert, 2022*). In this study, no significant correlation was found between hyperglycemia and initial testosterone levels. However, hyperglycemia was found to be associated with the lowest value of testosterone. The interactions between DM and testosterone are intricate. A study discovered that with every unit increase in fasting insulin, sex hormone-binding globulin decreased by 0.283 nmol/L, while bioavailable testosterone increased by 0.260 nmol/L (*Jiang et al., 2023*). Furthermore, testosterone can impact DM by participating in the DNA methylation process of suppressor of cytokine signaling-3 (SOCS3) (*Umapathysivam, Grossmann & Wittert, 2022*; *Jiang et al., 2023*). The increased risk of dyslipidemia is associated with inflammatory factors. Metabolism-related markers have both direct and indirect effects on dyslipidemia through the activation of inflammatory factors. In this study, it was observed that male patients with mPCa who had a triglyceride level of 1.7 mmol/L or higher took a longer time for their testosterone levels to decline to the lowest value. This suggests that the effectiveness of ADT may be partially counteracted by the metabolic and inflammatory side effects of periprostatic adipose tissue (*Lian et al., 2019*; *Mangiola et al., 2019*). There is ongoing debate regarding the connection between testosterone concentration and blood pressure. Some studies have suggested that low testosterone levels in men may be a predictor of hypertension, which is a risk factor for PCa (*Torkler et al., 2011*; *Stikbakke et al., 2022*). However, this study did not observe any effect of hypertension on testosterone levels before and after treatment, possibly due to limitations in sample size or population. *Lebdai et al. (2018)* discovered that the risk of adverse pathological features in PCa escalated as the composite metabolic score increased. A composite metabolic score of ≥4 points exhibited a noteworthy association with postoperative specimen ISUP grouping of ≥4 (OR = 1.9, P = 0.017) and positive resection margin (OR = 1.8, P = 0.007) (*Lebdai et al., 2018*). This study revealed a negative correlation between the composite metabolic score and both progression-free survival time and overall survival time in patients. In our study, we also observed that the lowest value of testosterone in the composite metabolic score group of 3 points or 4–5 was higher compared to the 0–2 group. Additionally, we found that the time required for testosterone levels to decline to the lowest value was also longer in the composite metabolic score group of 3 points or 4–5. However, we did not observe any significant difference in the lowest value of testosterone or the time it took for

testosterone levels to drop to the lowest value between the 3 points and 4–5 groups. While the interaction mechanism between MetS components and testosterone remains complex and not fully understood, our findings indicate that MetS components are associated with both testosterone levels and the time it takes for testosterone levels to decline in mPCa patients. The prognosis of mPCa appears to be closely linked to testosterone levels and decline time. It is possible that each MetS component contributes to the progression of mPCa by influencing factors related to testosterone. Moreover, this effect becomes more pronounced with a higher composite metabolism score.

## CONCLUSION

mPCa combined with MetS leads to a more aggressive condition, with a worse prognosis, with a higher lowest testosterone value after treatment and a longer time for it to reach the lowest value. MetS independently increases the risk of testosterone dropping to the lowest level for more than 6 months. The levels and timing of testosterone decline in mPCa patients are influenced by the components of MetS and the composite metabolic score. The higher the composite metabolic score the worse the prognosis of mPCa patients. Therefore, when monitoring testosterone levels in mPCa patients, it is important to consider the impact of MetS and its components, and adjust treatment strategies accordingly on an individual basis. This study is a single-center retrospective study with a limited sample size, which may introduce certain biases in the results. The positive results obtained for each component of MetS are also limited. Therefore, further verification is needed through multi-center, large-sample, and prospective studies. Additionally, it is necessary to conduct basic research to explore the intrinsic interaction mechanism between MetS and testosterone in PCa patients.

## ACKNOWLEDGEMENTS

We acknowledge the assistance of the data from the prostate cancer follow-up database of the Urology Center of the First Affiliated Hospital of Xinjiang Medical University. We also thank all the subject team members who participated in the study.

### Funding

This study was supported by the Key Program of Natural Science Foundation of Xinjiang Uygur Autonomous Region, China (2022D01D39), the Outstanding Youth Program of Natural Science Foundation of Xinjiang Uygur Autonomous Region, China (2023D01E05), the Regional Science Foundation of the National Natural Science Foundation of China (82360476), and the Program of Tianshan Talents of Xinjiang Uygur Autonomous Region, China (2022TSYCCX0026). The funders had no role in study design, data collection and analysis, decision to publish, or preparation of the manuscript.

## Grant Disclosures

The following grant information was disclosed by the authors:

Key Program of Natural Science Foundation of Xinjiang Uygur Autonomous Region, China: 2022D01D39.

Outstanding Youth Program of Natural Science Foundation of Xinjiang Uygur Autonomous Region, China: 2023D01E05.

Regional Science Foundation of the National Natural Science Foundation of China: 82360476.

Program of Tianshan Talents of Xinjiang Uygur Autonomous Region, China: 2022TSYCCX0026.

## Competing Interests

The authors declare there are no competing interests.

## Author Contributions

- Tao Zhuo conceived and designed the experiments, performed the experiments, analyzed the data, prepared figures and/or tables, authored or reviewed drafts of the article, and approved the final draft.
- Xiangyue Yao performed the experiments, analyzed the data, prepared figures and/or tables, authored or reviewed drafts of the article, and approved the final draft.
- Yujie Mei performed the experiments, analyzed the data, authored or reviewed drafts of the article, and approved the final draft.
- Hudie Yang performed the experiments, analyzed the data, authored or reviewed drafts of the article, and approved the final draft.
- Abudukeyoumu Maimaitiyiming performed the experiments, prepared figures and/or tables, and approved the final draft.
- Xin Huang performed the experiments, prepared figures and/or tables, and approved the final draft.
- Zhuang Lei performed the experiments, prepared figures and/or tables, and approved the final draft.
- Yujie Wang analyzed the data, prepared figures and/or tables, and approved the final draft.
- Ning Tao conceived and designed the experiments, authored or reviewed drafts of the article, and approved the final draft.
- Hengqing An conceived and designed the experiments, authored or reviewed drafts of the article, and approved the final draft.

## Human Ethics

The following information was supplied relating to ethical approvals (*i.e.*, approving body and any reference numbers):

Ethical Approval was obtained from Medical Ethics Committee of the First Affiliated Hospital of Xinjiang Medical University (No. 20220308-166).

## Data Availability

The raw data is available in the Supplementary File.

## Supplemental Information

Supplemental information for this article can be found online at http://dx.doi.org/10.7717/peerj.17823#supplemental-information.

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
