# Peer review of "Effect of metabolic syndrome on testosterone levels in patients with metastatic prostate cancer: a real-world retrospective study"

_PeerJ, doi:10.7717/peerj.17823_

## Round 0.1 · original submission · Major Revisions

· Academic Editor

Major Revisions

Dear Authors,

The study entitled “Effect of Metabolic Syndrome on Testosterone in Patients with Metastatic Prostate Cancer” presents compelling findings using a robust methodological approach. However, several important points require clarification within the manuscript. Following the peer review process and an additional review from a third reviewer, we have determined that your article holds significant potential for publication in PeerJ. Nonetheless, the reviewers have requested substantial revisions.

We look forward to your resubmission.

Sincerely,

·

Basic reporting

Overview and general recommendation:
This manuscript investigates the impact of metabolic syndrome on testosterone levels in patients diagnosed with prostate cancer. The results demonstrated that the severity of prostate cancer is greater in patients with metabolic syndrome (higher tumor burden, greater percentage of stage ≥ 4, and Gleason score ≥ 8). There was no difference in initial testosterone levels between groups but the treatment-induced reduction on testosterone levels was slower and less intense in the MetS group. The results are interesting and relevant but a few shortcomings pertaining to the study are listed in the present and following sections. Besides, I suggest that the authors carefully proof-read the manuscript to eliminate some simplifications often common in “hospital language” to render the text more elegant. For example, “effect of metabolic syndrome on testosterone in patients…” could be changed to testosterone levels/concentration or any best suited word. The raw data spreadsheet should be fully translated to English to allow a more thorough evaluation of the findings by the audience. Finally, the authors should also consider altering the title to include information about the findings of the study.
Minor comments:
- Include some background information in the abstract.
- Substitute “higher lowest values” for clarity of information.
- Include reference for sentences on page 7, lines 56-57 and 65-66.
- Detail the indicators and units on page 8, lines 93-95 (for example, initial PSA, initial testosterone…).
- Include reference for sentence on page 12, lines 197-198 and 210-211.

Experimental design

1. Some details about the androgen deprivation therapy (ADT) should be included in the text. Did all patients receive the same treatment regimen for prostate cancer? Were there any differences in the metabolic syndrome management? Could the results have been influenced by any drug-related interference on the parameters analyzed in the study?
2. Why did the authors choose 6 months as a time reference for the data analysis represented in table 3? Please explain the rationale.
3. On page 7, lines 56-57, the authors mentioned the survival rate for prostate cancer. Later on page 8, line 98, it is possible to understand that data from deceased patients were included in the analysis. Considering this and the time frame (2010-2022), it would be interesting to revisit the clinical files and evaluate the survival rate in both groups.
4. On page 7, lines 72-73, the authors stated that “Previous studies have demonstrated that MetS has an impact on PSA levels in patients with PCa”; however, there was no change in PSA levels in the MetS group compared to Non-MetS subjects. How could this be explained?
5. Considering that there was no significant correlation between hyperglycemia and testosterone levels in the statistical analysis, the authors should rewrite the sentence on page 14, lines 259-261.

Validity of the findings

The conclusion is well supported by the results of the study. However, the objectives listed on the text are broader than in the abstract. The authors should consider substituting the version on the abstract to encompass all the analysis performed in the study.

Reviewer 3 ·

Basic reporting

The study titled "The impact of metabolic syndrome on testosterone levels in patients diagnosed with prostate cancer" aimed to investigate how factors related to metabolic syndrome affect testosterone levels before and after treatment, as well as the time it takes for testosterone to decline in patients with metastatic prostate cancer.

The main findings of this study were:

The severity of prostate cancer (indicated by a higher occurrence of high tumor burden) is greater in patients with metabolic syndrome, suggesting that metastatic prostate cancer combined with metabolic syndrome is more aggressive.

The reduction in testosterone levels induced by treatment was slower and less pronounced in the metabolic syndrome group.

Recommendations for improving the manuscript:

The raw data spreadsheet should be fully translated into English.

Authors must review and include all missing references in the manuscript.

Indicators and units should be to better detailed in certain parts of the manuscript.

Experimental design

The experimental design of this study is adequate. However, some important points must be clarified in the manuscript.

Major revisions:

Did the authors evaluate the medications used by study participants? Can it be stated with certainty that the results of this study were not influenced by any interference from the medication?

Did all patients in the study receive the same treatment regimen for prostate cancer?

The authors mention that previous studies have shown an impact of metabolic syndrome on PSA levels in patients with metastatic prostate cancer. However, your study did not observe changes in PSA levels between groups. I suggest that the authors revise this part of the text.

Validity of the findings

These results are both interesting and highly relevant. The conclusion is well supported by the results of the study.

Additional comments

No comments

---

## Round 0.2 · accepted · Accept

· Academic Editor

Accept

Dear Author,

Congratulations! After your diligent work addressing the reviewers' comments, I am pleased to inform you that your manuscript has been accepted for publication in PeerJ. This version is more concise and formal, enhancing clarity and flow.

Reviewer 3 ·

Basic reporting

The authors have adequately addressed all my comments and revised it accordingly.
The manuscript is significantly improved upon the readability and clarity of the manuscript. It is well-structured, designed, and referenced. Therefore, I have no further comments.

Experimental design

The authors have adequately addressed all my comments and revised it accordingly. Therefore, I have no further comments.

Validity of the findings

I have no further comments.

Additional comments

I appreciate the point-by-point response to my review and I want to congratulate the authors for their hard work.